# Topological Anderson Insulator in Cation-Disordered Cu_2_ZnSnS_4_

**DOI:** 10.3390/nano11102595

**Published:** 2021-10-01

**Authors:** Binayak Mukherjee, Eleonora Isotta, Carlo Fanciulli, Narges Ataollahi, Paolo Scardi

**Affiliations:** 1Department of Civil, Environmental and Mechanical Engineering, University of Trento, Via Mesiano 77, 38123 Trento, Italy; eleonora.isotta@unitn.it (E.I.); narges.ataollahi@unitn.it (N.A.); 2National Research Council of Italy, Institute of Condensed Matter Chemistry and Technologies for Energy (CNR-ICMATE), Lecco Unit, Via Previati 1/E, 23900 Lecco, Italy; carlo.fanciulli@cnr.it

**Keywords:** quaternary chalcogenides, topological insulators, disordered systems, kesterite, thermoelectrics

## Abstract

We present the first candidate for the realization of a disorder-induced Topological Anderson Insulator in a real material system. High-energy reactive mechanical alloying produces a polymorph of Cu_2_ZnSnS_4_ with high cation disorder. Density functional theory calculations show an inverted ordering of bands at the Brillouin zone center for this polymorph, which is in contrast to its ordered phase. Adiabatic continuity arguments establish that this disordered Cu_2_ZnSnS_4_ can be connected to the closely related Cu_2_ZnSnSe_4_, which was previously predicted to be a 3D topological insulator, while band structure calculations with a slab geometry reveal the presence of robust surface states. This evidence makes a strong case in favor of a novel topological phase. As such, the study opens up a window to understanding and potentially exploiting topological behavior in a rich class of easily-synthesized multinary, disordered compounds.

## 1. Introduction

Topologically non-trivial materials present a novel and exciting field of research in condensed matter [1]. They are valued both for their importance to fundamental science as exotic states of quantum matter as well as their inherent potential for application in new and future technologies including thermoelectrics [2,3,4], spintronics [5,6,7], and quantum computation [5,6,8]. Starting with the discovery of the Quantum Hall Effect (QHE) by von Klitzing et al. [9], this class of materials has grown to include many candidates in 2-, 3-, and higher dimensional systems, a growing (albeit still small) fraction of which have been experimentally realized. Three-dimensional (3D) topological insulators (TIs) present a sub-class of these exotic materials. They may generally be described as hosting insulating bands in the bulk with band inversion at high-symmetry points, coupled with symmetry-protected gapless surface states [10]. In the absence of symmetry breaking, these surface states support high-mobility electron transport along specific directions on the surface, without backscattering. Large spin-orbit coupling (SOC) was originally understood to be driving the topologically non-trivial behavior [11,12,13,14,15]. Subsequently, Fu [16] demonstrated that topological surface states can also be protected by crystalline symmetries in the absence of SOC (topological crystalline insulators). This allows for topologically non-trivial materials with a weak SOC [17].

The possibility of TIs in the quaternary chalcogenide class has been investigated by Chen et al. [18], using density functional theory (DFT) band structures. They showed that HgTe, a 3D semimetal with the zinc-blende structure, may be transformed into a TI by introducing a strong crystal field splitting (Δ*_CF_*). This can be achieved either by epitaxial straining or by substituting two group-II Hg ions with one group-I ion and one group-III ion. The latter approach results in ordered I-III-VI_2_ chalcopyrites. Subsequently, by replacing two group-III cations with one group-I and one group-II cation, the I_2_–II–IV–VI_4_ chalcogenides are obtained. The non-trivial band gap of these materials could be increased further via a simultaneous increase in Δ*_CF_* and the band-inversion strength (BIS). In a contemporaneous study, Wang et al. [19] performed a DFT-based screening of several ternary famatinite and quaternary chalcogenides for TIs and were able to identify several naturally occurring, Cu-based 3D TIs. Unsurprisingly given its weak SOC, Cu_2_ZnSnS_4_ (CZTS) was found to be topologically trivial, although the authors showed that it could be ‘transformed’ into a TI by changing the atomic number of the cations, which was manifested as a doping effect evolving toward the ternary TI Cu_3_SbS_4_.

Topological insulators, including the multinary compounds mentioned above, are generally known to host a bulk band gap coupled to gapless surface states, which are robust to weak levels of disorder. Several recent studies have highlighted how TI behavior can exist in aperiodic systems such as quasicrystals [20] and can persist in systems with bulk defects such as grain boundaries and vacancies below a certain threshold [21,22]. Nevertheless, sufficiently strong disorder is expected to close the bulk gap and destroy all topological features [10,22]. In light of this, a surprising prediction was made by Li et al. [23], who claimed that adding disorder to otherwise trivial systems can lead to the emergence of topological behavior. The authors showed that disorder-induced Anderson localization may lead to a renormalization of the topological mass of the charge carriers via the band structure, causing a transition from a topologically trivial phase to a TI. This gives rise to the so-called Topological Anderson Insulator (TAI) phase. TAIs have been theoretically shown to be feasible by introducing disorder into trivial 3D systems close to a topological phase [24,25]. TAI behavior was first demonstrated by Meier et al. [26] using quantum simulations in a metamaterial consisting of a 1D chain of ultracold rubidium atoms. Subsequently, in a recent article by Nakajima et al. [27], using a Thouless pump realized with ultracold ytterbium atoms on a dynamical optical lattice, the authors demonstrate a disorder-induced pumping, with a topologically trivial phase in the clean limit driven to a non-trivial phase due to quasi-periodic disorder. However, to date, evidence of TAI phases in a real material remains conspicuously absent.

Crucially, the quaternary chalcogenides screened in the aforementioned studies including CZTS are all ordered tetragonal structures. However, it is known that CZTS crystallizes in multiple polymorphs. In a recent study, Isotta et al. [28] demonstrated remarkably improved thermoelectric properties in a cubic polymorph of CZTS with complete cation disorder. This polymorph, which was synthesized using high-energy reactive mechanical alloying (ball-milling), shows a simultaneous improvement in both Seebeck coefficient and electrical conductivity, as well as a lower thermal conductivity compared to the ordered tetragonal polymorph. Using the DFT band structure calculations, we argue that introducing full cation disorder in CZTS drives it into a TAI phase; experimental measurements of electrical resistivity and carrier mobility are in agreement with a surface contribution to transport, which such a phase is expected to host. As such, we present the first concrete prediction of a TAI in a material, opening up myriad possibilities to investigate topologically non-trivial behavior in disordered quaternary compounds and their potential effect on thermoelectric performance.

## 2. Computational Methodology

The ab initio electronic structure calculations have been performed using the plane wave basis set implemented in the Vienna ab initio simulation package (VASP), version 5.4.4, Vasp Software GmBH [29,30]. The electron-exchange correlation functional was approximated using the Perdew−Burke−Ernzerhof (PBE) [31] form of the generalized gradient approximation (GGA). It should be noted that the GGA tends to underestimate the band gap for most compounds, which may be corrected using computationally expensive hybrid functionals [32]. However, a hybrid Hartree-Fock/DFT study [33] has established that the band topology for both computational schemes (hybrid and PBE) are very similar for CZTS. The hybrid functional only shifts the conduction bands to a higher energy, justifying our use of the standard, computationally inexpensive PBE functional. In order to preclude a spurious negative band gap, we have also performed a single-point calculation with the HSE06 functional, with a 25% contribution from the exact Fock exchange energy. All calculations were performed with an energy cutoff of 450 eV and Gaussian charge smearing in the order of 0.01 eV. Calculations for CZTS were performed both with and without spin-orbit coupling, while calculations for CZTSe were made only with SOC. The geometry was optimized with a 2 × 2 × 2 Monkhorst Pack (MP) Γ-centered k-mesh for 64 atom supercells until Hellman–Feynman forces on each atom were converged to below 0.01 eV/Å. SCF calculations were made with a similar 4 × 4 × 4 k-mesh, with electronic degrees of freedom relaxed until the change in the total free energy and energy eigenvalues were both smaller than 10^−6^ eV. Calculations for surface states were performed within a surface slab geometry, with a 10 Å vacuum layer in the Z-direction to minimize the interaction between periodic copies. Only the top three layers of the slab were allowed to relax, with lower layers held fixed. Geometry optimization for the surface slab was made with a 2 × 2 × 1 MP k-mesh, while SCF calculations used a 4 × 4 × 2 mesh. For band structure calculations, the high-symmetry path in the Brillouin zone was obtained using SeekPath [34]. VESTA [35] was used for visualizing atomic geometries. The disordered geometries are constructed by generating a peseudorandom number between 1 and 32 and assigning each cation to the corresponding serially numbered cation site.

## 3. Results and Discussion

### 3.1. Band Inversion in the Bulk

CZTS is a quaternary chalcogenide compound extensively investigated for its potential applications, primarily in photovoltaics [36,37,38,39,40,41], and recently thermoelectrics [28,42,43,44,45,46,47]. The most ubiquitous polymorph of CZTS is the kesterite structure (Figure 1a), which crystallizes in the tetragonal *I-4* space group. The structure may be described as alternating layers of cations and sulfurs, with a further alternation in the cation layers, which are either composed of Cu and Zn or Cu and Sn. Above 533 K, this *I-4* structure undergoes the so-called order–disorder transition [48] into a tetragonal *I-42m* phase. In this structure, the disorder is manifested through an in-plane randomization of the cations in the Cu-Zn layer (Figure 1b). This disorder induces a narrowing of the band gap compared to the ordered tetragonal polymorph (Figure 1d,e), while the increase in global symmetry introduces a three-fold degeneracy at the center of the irreducible Brillouin zone (Γ-point) in the valence band maximum (Figure 1e). In a recent article, Isotta et al. [28] have presented the synthesis of a novel polymorph of CZTS, this time with full cation disorder. This is manifested as a complete randomization of atoms in the cation position (Figure 1c). This polymorph was found to crystallize in the cubic zinc-blende/sphalerite structure with space group *F-43m* and remain stable up to 673K when it transitions to the tetragonal polymorph. Electronic structure calculations revealed the presence of significant inhomogenous bonding. This removes the three-fold degeneracy present in the bands (Figure 1f) of the disordered tetragonal polymorph, via strong crystal field splitting, while opening up the band gap somewhat (see Appendix A).

A common feature of all three polymorphs of CZTS is that the states in the valence band maximum (VBM) are dominated by the Cu-d electrons, while those in the conduction band minimum (CBM) are mainly derived from S-p orbitals (Figure 1d–f). However, upon closer inspection of the projected bands for cubic CZTS, we observe that the order of the bands is reversed at and around the Γ-point, with an inversion in the Cu-d and S-p orbitals (Figure 1f) (see Appendix A for discussion on band gap).True disorder in ionic positions is of course rather difficult to simulate within the size constraints of a DFT supercell with periodic boundary conditions, which impose a long-range order on the system. In order to ensure that the band inversion is not an accidental artifact but rather a property of the system, we have calculated the band structure for a further nine different configurations of cubic CZTS (Appendix A), with Cu, Zn, and Sn ions randomly assigned to each cation position, while maintaining the Cu_2_ZnSnS_4_ stoichiometry (see Appendix A) for energies of each configuration. The lowest energy configuration is shown in Figure 1c,f. Additionally, in order to discard the possibility that an underestimation of the band gap by the PBE functional leads to a spurious band inversion, a single-point calculation was performed using the computationally expensive HSE06 hybrid functional. This confirms a negative band gap in the order of −0.12 eV.

The features of the bands are necessarily somewhat different from each other. This is because each configuration of disorder generates a different kind of inhomogeneity in the charge distribution, leading to different levels of crystal field splitting. Crucially, however, band inversion is present in every case, which is coupled in most cases with an anti-crossing (camel’s back) feature at VBM and CBM. In fact, the band-inversion strength, as defined by the energy difference between the lowest inverted S-p level in the valence band and the highest Cu-d level, is found to be reasonably positively correlated with Δ*_CF_*, with a Pearson’s r value of 0.84 (Figure 2a). While SOC is known to play a driving role in most topologically non-trivial systems, previous studies [49] have shown that it is negligible for tetragonal CZTS. We confirm that this remains the case in the cubic polymorph. Including SOC in the calculation does not significantly alter the nature of the bands (Appendix A) at the valence and conduction band extrema compared to the bands obtained without SOC (Appendix A). Band inversion remains intact in both cases, suggesting that SOC might not be the main feature driving the system into a TI phase.

Instead, we assert that it is the large Δ*_CF_* that causes the inversion and opens up a non-trivial band gap. This is in line with the arguments proposed by Chen et al. [18] in the case of strained HgTe and tetragonal ternary chalcopyrites and quaternary chalcogenides. Here, the Δ*_CF_* is a result of inhomogeneous bonding, arising from the full cation disorder in cubic CZTS. In light of this disorder-induced topological transition, we propose cubic CZTS as a candidate Topological Anderson Insulator.

It is well known [50] that an inverted band structure corresponds to a negative (topological) effective fermion mass, *m*. In the TAI phase, Groth et al. [24] have demonstrated that this inversion is obtained as a result of elastic scattering from a disorder potential, which leads to states with a definite momentum decaying exponentially as a function of space and time. When the effective Hamiltonian of the disordered system acts on the exponentially decaying state, it adds a negative correction *δm* to the effective mass. This renormalized topological mass *m’ = m + δm* can have a sign that is opposite that of the bare mass *m*, corresponding to a band inversion. The low energy Hamiltonian *H* of a general 3D topological Anderson insulator was written by Guo et al. [25] as,
(1)$$H=H_{0}+\underset{j}{\sum }U_{j}\Psi_{j}^{†}\Psi_{j}$$
where $H_{0}$ is the Hamiltonian of the ordered (trivial) system, $\Psi_{j}$ is the overall wave function at the *j*-th lattice site, and $U_{j}$ is the on-site disorder or Anderson potential. By definition, the Anderson potential must vary randomly within the crystal lattice, and it will correspond to a random component in the local potential energy in addition to the periodic component due to the crystal lattice. 

In order to compare the differences in the potential for the different polymorphs, we compute the local potential energy along the X-, Y-, and Z-directions in the CZTS supercells. It is evident from Figure 2b,c (and Appendix A) that the potential in the ordered tetragonal polymorph (black curve) exhibits a highly periodic nature. Instead, the cubic structure (red curve) deviates significantly from this periodicity. This is quite similar to the maximum quasi-periodic disorder of the Thouless pump reported by Nakajima et al. [27], which drives a trivial phase into a non-trivial one. As such, we assert that the potential in the disordered polymorphs can be safely approximated to be the potential of the ordered structure plus a modifying potential due to disorder, which is in the spirit of Equation (1). Then, this modifying term is given by the difference between the ordered and disordered potentials, as seen in Figure 2b,c (and Appendix A).

Critically, it has been demonstrated [51] that bond disorder, which adds random hopping terms to the Hamiltonian, and is present in many material systems, cannot drive a system into the TAI phase. As such, the bonding inhomogeneity prevalent in disordered CZTS cannot be held responsible for the non-trivial nature of the system. Instead, it is an independent by-product of the same random on-site cation disorder potential, which also gives rise to the TAI behavior. This is evident from Figure 2b,c, which show the random disorder potential in the x- and y-directions, respectively. These directions are not in fact the bonding directions in CZTS, thus putting our results in agreement with those of Song et al. [51].

Instead, Girschik et al. [52] have suggested that any long-range correlations in the disorder potential might lead to a strong suppression of the TAI phase. Such correlations can be reasonably precluded from our system by considering the global nature of the disorder in CZTS. This constitutes a total randomization of atomic species in the cation lattice sites. Thus, the nature of the disorder prevents long-range correlations, instead promoting short-range, random variations of the local potential and allowing for the TAI phase to manifest.

Then, it is clear that the modifying potential is a highly random and aperiodic short-range onsite potential. This makes it a suitable candidate for the on-site disorder potential in the theory of TAIs. Given the previous predictions of the closely related Cu_2_ZnSnSe_4_ as a TI [19], and the presence of a strong Anderson potential in the cubic polymorph, we put forward that ordered CZTS is driven into the disordered TAI phase. This is the result of introducing a high level of cation disorder, such as can be achieved through high-energy ball milling.

### 3.2. Adiabatic Continuity 

The presence of band inversion in the bulk is considered a necessary but not sufficient condition for the presence of a topological insulator phase [53]. To this end, adiabatic continuity arguments have emerged as a powerful tool to characterize the topological nature of materials through ab initio calculations, and they have been used to predict new TI phases [54,55,56,57,58]. The process involves connecting a known topological material to a new structure through a series of adiabatic changes. These include straining the crystalline lattice, tuning the strength of the SOC, and modifying the nuclear charge of constituent atoms within the constraint of overall charge neutrality [53]. If the Hamiltonian of this new system can be adiabatically connected to that of the known TI via some combination of the aforementioned ways without inducing a band inversion or a closing of the gap, the new material can be considered to be topologically equivalent to the known material and thus also a TI. Previous studies [18,19] have adiabatically connected quaternary chalcogenides to the known TI HgTe (strained) via both ternary famatinites and chalcopyrites. Of these compounds, the closest to our present case is the proposed TI [19] Cu_2_ZnSnSe_4_ (CZTSe) with an *I-42m* tetragonal stannite structure.

Starting from this structure, we are able to transition to a fully disordered CZTSe with a cubic F-43m lattice by introducing randomization. This is done by interchanging the coordinates of a single pair of cations at a time. Given that both stoichiometry and charge remain conserved overall, such a transition corresponds to an adiabatic change of the total Hamiltonian of the system (see Appendix A for energies of the intermediate configurations in the transition). Subsequently, we replace selenium ions with sulfur in the anion position, thereby transitioning into our cubic CZTS. This, once more, is an adiabatic transition given that both S and Se are group IV elements with identical s2p4 outer-shell electronic configurations. The extra contribution of Se is only through fully occupied core levels that lie well away from the Fermi energy. As Figure 3 demonstrates, this entire transition can be made without closing the inverted band gap at the Γ-point. Thus, we can conclude that the disordered cubic polymorph of CZTS is in fact topologically connected to the previously predicted TI CZTSe in the adiabatic limit (see Appendix A for band structures of the entire transition). It should be noted that the high degree of cation disorder implicitly increases the global symmetry of CZTS from tetragonal to cubic with two interpenetrating sub-lattices of cations and sulfur anions. This structure lies in the same F-43m space group as HgTe, which is the parent compound of this family of adiabatically connected 3D TIs.

### 3.3. Topological Surface States

Further evidence to verify the topologically non-trivial nature of a material can be obtained by characterizing the topologically protected gapless surface states, which are guaranteed through bulk-boundary correspondence. The calculation of these states from first principles is computationally demanding, and the results can be distorted by spurious gaps due to interactions between periodic copies [53]. In order to reduce these artefacts, we calculate the surface states for a 001 surface with sulfur termination within a slab geometry, with a large (10 Å) vacuum introduced in the Z-direction (see Appendix A for a simulation cell). Other than the expected quasi-gapless state at the Γ-point, our calculations point to a further two states at the high-symmetry R and V points (Figure 4a). This is in agreement with the requirement for an odd number of Dirac points. It must be mentioned that these states tend more toward a parabolic curvature rather than the well-known linearly-dispersing Dirac cones. This “quadratic band touching” was predicted by Fu [16] using a tight-binding model for spinless fermions. In real materials, this corresponds to systems with weak SOC. These so-called “Schroedinger paraboloids” have recently been reported in novel topological semi-metals, namely the Weyl semi-metal candidate SrSi_2_ [59] and the so-called Schroedinger semimetal Be_2_P_3_N [60]. Both these materials, interestingly, show weak SOC similar to cubic CZTS. 

Topological surface states can be sharply distinguished from well-known trivial surface states in semiconductors/insulators. The latter are less robust and can be removed via surface deformation [53]. Interestingly, we find that the topologically trivial ordered tetragonal CZTS hosts such surface states at the E and C_2_ high-symmetry points on the 001 surface (Figure 4d). In order to test for robustness, we have deformed the surface by applying a small (1%) in-plane expansive strain (Figure 4e) as well as simply removing a single S atom from the surface layer (Figure 4f). This leads to an opening of the gap in ordered tetragonal CZTS, causing the trivial gapless states to vanish. In comparison, the surface states in the disordered cubic polymorph are found to be significantly resilient (Figure 4c,d) to identical surface treatments, confirming that these states are in fact topologically protected. These robust surface states are expected to support quasi-metallic surface transport. This would contribute significantly to the improved conductivity observed in disordered cubic CZTS [28].

### 3.4. Discussion 

In general, the evidence presented above, while strongly suggesting that disordered cubic CZTS behaves as a TAI, cannot be considered to be conclusive. Further investigation, both theoretical and experimental, is part of ongoing research. Theoretical calculations using tight-binding models and effective Hamiltonians can be used to calculate the Berry phase and topological invariant, in order to provide a more fundamental understanding of the topologically non-trivial behavior of the material. On the experimental side, conclusive evidence for topological surface states can be obtained with angle resolved photoemission spectroscopy (ARPES). However, the nano-polycrystalline nature of our samples makes ARPES unfeasible. Preliminary electrical measurements indicate an inverse relation between grain size and carrier mobility, suggesting a strong surface contribution (see Appendix A). However, further transport measurements at ultra-low temperatures and in the presence of magnetic fields might be used to better characterize the nature of the surface states.

## 4. Conclusions 

In the present article, we propose a possible candidate for a disorder-induced TI material, the so-called Topological Anderson Insulator. High-energy reactive ball milling has recently been used to produce a low-temperature disordered cubic (*F-43m*) phase of the quaternary chalcogenide Cu_2_ZnSnS_4_, with complete randomization in the cation positions. DFT calculations show that this novel disordered polymorph has an inverted band order in the conduction and valence band extrema at and close to the Brillouin zone center, in contrast to the trivial bands of the ordered tetragonal (*I-4*) counterpart. Furthermore, the band-structure of this phase can be connected adiabatically to the bands for ordered tetragonal (*I-42m*) Cu_2_ZnSnSe_4_, which is known to be a 3D TI, without closing the inverted band gap. Surface slab calculations reveal the presence of an odd number (three) of quasi-gapless surface states, which are remarkably robust to surface deformation such as strain and defects. This is in sharp contrast to the fragile surface states in ordered CZTS. 

The DFT calculations presented here offer a strong argument in favor of this novel topological phase in ball-milled cubic CZTS with full cation disorder. While not claiming conclusive proof, this work opens up a significant possibility for topological matter in the realm of multinary, disordered compounds, where topological behavior is only partially understood. Such materials, which are easily and cheaply synthesized in comparison to perfect crystals for traditional TIs, open up diverse possibilities not just for fundamental research but also the application of topological properties, particularly in the area of thermoelectrics.

## Figures and Tables

**Figure 1 nanomaterials-11-02595-f001:**
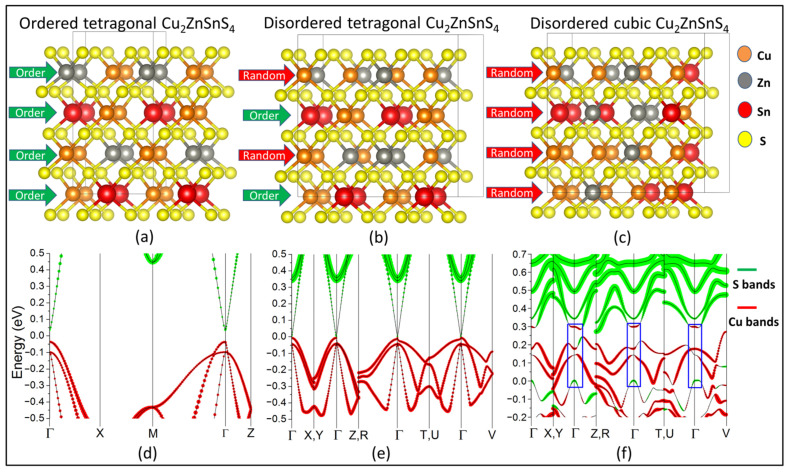
Crystal geometries and orbital-projected bands for the different polymorphs of CZTS: (**a**) the geometry of ordered tetragonal CZTS; (**b**) a supercell of disordered tetragonal CZTS; (**c**) a supercell of disordered cubic CZTS; Orange balls refer to copper ions, red balls refer to tin, gray balls refer to zinc, and yellow balls refer to sulfur; green arrows identify the ordered layers, while red arrows show the layers with cation randomization; (**d**) the bands for ordered tetragonal CZTS; (**e**) the bands for disordered tetragonal CZTS; and (**f**) the bands for cubic CZTS; green circles correspond to the dominant contribution from sulfur-p orbitals, while red circles correspond to the contribution from copper-d orbitals; the blue box highlights the region of band inversion in cubic CZTS.

**Figure 2 nanomaterials-11-02595-f002:**
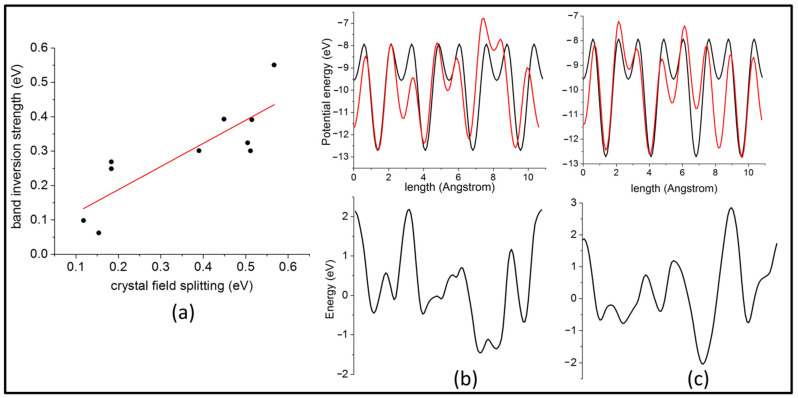
(**a**) Correlation between band inversion strength and crystal field splitting; (**b**) the local potential in ordered tetragonal (black line) and cubic (red line) CZTS along the x-direction (above) and the difference between the two (below); (**c**) the local potential in ordered tetragonal (black line) and cubic (red line) CZTS along the y-direction (above) and the difference between the two (below).

**Figure 3 nanomaterials-11-02595-f003:**
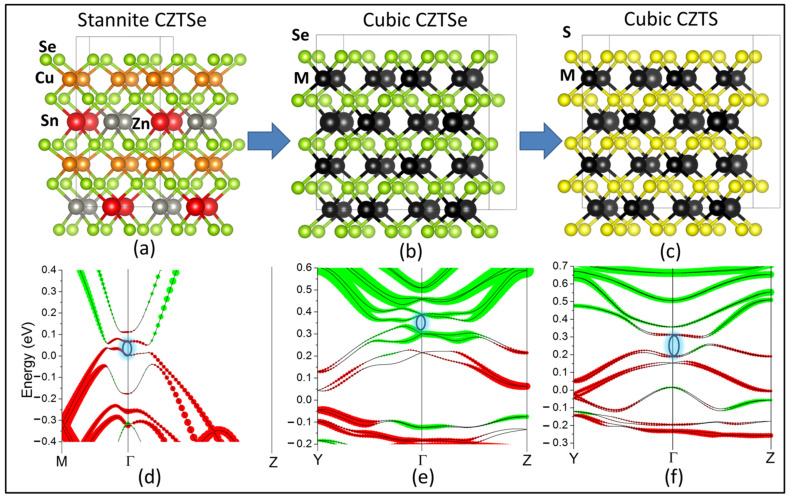
Adiabatic continuity between stannite CZTSe and cubic CZTS: (**a**) the crystal geometry of stannite CZTSe, (**b**) the geometry of cubic CZTSe, (**c**) the geometry of cubic CZTS); orange, red, gray, and yellow circles have the same meaning as in Figure 1, green circles refer to selenium ions, black circles refer to a randomized cation; (**d**) the bands of stannite CZTSe, (**e**) the bands of cubic CZTSe, (**f**) the bands of cubic CZTS; red circles correspond to the dominant contribution from Cu-d orbitals, green circles correspond to the dominant contribution from anion-p orbitals; the black arrow shows the open non-trivial band gap.

**Figure 4 nanomaterials-11-02595-f004:**
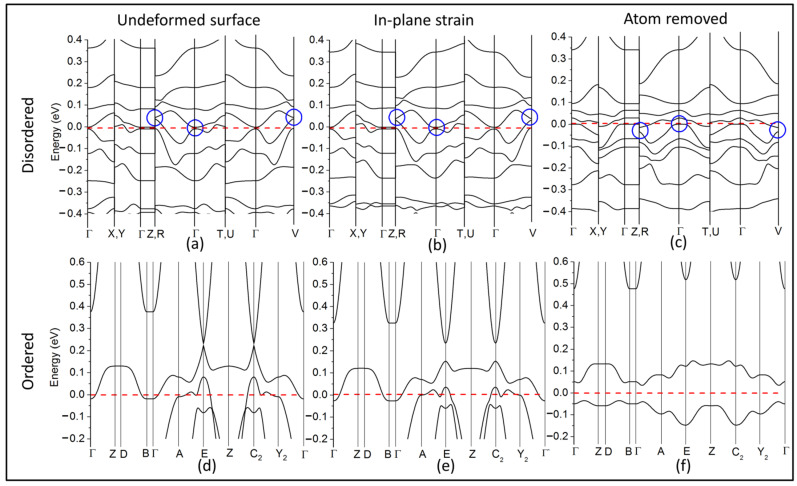
Surface states: (**a**) the surface band structure of cubic CZTS, (**b**) the corresponding surface bands under 2% expansive strain in the XY plane, (**c**) the corresponding surface bands with one sulfur atom removed from the surface; blue circles mark the gapless surface states; (**d**) the surface band structure of ordered tetragonal CZTS; (**e**) the corresponding surface bands under 1% expansive strain in the XY-plane, (**f**) the corresponding surface bands with one sulfur atom removed from the surface.

## Data Availability

The data that support the findings of this study are available from the corresponding authors upon reasonable request.

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
