# Peer review of "Topological Anderson Insulator in Cation-Disordered Cu2ZnSnS4"

_nanomaterials, 2021, doi:10.3390/nano11102595_

Round 1
Reviewer 1 Report
The authors mentioned that they made significant modifications, e.g., completely removing impedance spectroscopy results as well as adding transport results. This is good. However, I have a major concern for the transport data the authors have provided, and therefore, I cannot recommend publication in Nanomaterials. The following are details of the concern.
There are discrepancies between the data in the original manuscript and those in the revised version. For example, the temperature dependence of the transport data in Fig. 5c in the revised version made me puzzled, i.e., it looks totally different from the data provided in Fig. 5a in the original one.
- The original resistivity data in Fig. 5a demonstrate a clear temperature dependence, while the revised conductivity data in Fig. 5c don’t.
- The resistivity for a “just pressed CZTS sample” is 100 mΩ•m (=0.1 Ω•m) = 10 Ω•cm. Then, conductivity must be an order of 0.1 S/cm, however, the new data in Fig. 5c shows that the conductivity is over 50 S/cm (about 500 times larger). Why is the magnitude of the resistivity (or conductivity) so different between two? The authors should double check the data/analysis and reproducibility of the data, which is a key for publication.
The authors mentioned in lines 359-360 as “thermal treatments typically lead to higher density, improved connectivity between adjacent domains, and a reduction in carrier scattering from the grain boundary”. I understand that sintering grains (i.e., thermal treatments) is performed to promote some grain growths from Fig 5a and 5b, and Fig, S6. However, the TEM images in Fig. 5 give me an impression that the density for just pressed sample looks higher than that for sintered assuming that the definition of the density is the ratio between filled area by grains and empty (porous) area in the images. I would agree that the TOTAL connecting area where adjacent grains/domains contact has something to do with the connectivity with respect to conductivity, though it does not look so easy to estimate the area from a 2D image like those in Fig. 5. Have the authors quantitatively estimated it? Otherwise, “improved connectivity” sounds a speculation as it is not obvious.
Furthermore, in general, it is not so simple to precisely estimate the carrier concentration and mobility of porous samples like what the authors have from the Hall data because the orientations and distributions of current flows inside the sample are not trivial against the applied magnetic field due to the presence of randomly oriented chains of grains. I imagined that the samples are more packed, which turns out to be incorrect from the TEM images in the revised version. The Hall data may be also affected by how/where the electrode contacts were made in this case. The authors should provide such information so that readers can judge how reliable the transport measurements were.
Finally, I wonder if topological surface states (TSS) of CZTS grain should vanish at the grain boundaries? When two topologically nontrivial materials are attached, then the original TSSs at the interface should vanish as TSSs could emerge at interface where trivial and nontrivial materials meet. I also wonder if the gapless (metallic) TSSs would be gapped (not metallic) due to interference of the wavefunctions between two sides of the surface of a grain when the size of grains becomes so small (smaller than several nm). If this is the case, what are observed have nothing to do with topological states. It may be prudent to consider other possibilities rather than a contribution from TSSs in this case to provide a conductivity difference between just pressed and sintered samples.
Author Response
There are discrepancies between the data in the original manuscript and those in the revised version. For example, the temperature dependence of the transport data in Fig. 5c in the revised version made me puzzled, i.e., it looks totally different from the data provided in Fig. 5a in the original one. The original resistivity data in Fig. 5a demonstrate a clear temperature dependence, while the revised conductivity data in Fig. 5c don’t.
Response: The transport data provided in the present version of the manuscript are indeed different from those in the previous version. The ‘discrepancies’ mentioned here have actually been preempted and clearly addressed in our previous response to the esteemed reviewer: “These samples, produced by a slightly different synthesis technique (using brass rather tungsten carbide ball for ball milling), provide lower resistivity values, making Hall effect measurements more convenient. For the sake of consistency, we have completely replaced the previous set of resistivity measurements with the new set.” If one compares the old and new data, it is clear that there is an order of magnitude increase of conductivity in the latter.
With respect to the temperature dependence, a close observation of the conductivity data provided in the new Fig 5c reveals a small but perceptible increase with temperature, in agreement with the downward trend in the resistivity data in Fig 5a in the original version. The apparent visual difference between the trends is of course due simply to the difference in scales on the y-axis. The order-of-magnitude higher values in the new data make the changes with temperature higher to discern; nevertheless, the old and new data in fact have the same trend.
The resistivity for a “just pressed CZTS sample” is 100 mΩ•m (=0.1 Ω•m) = 10 Ω•cm. Then, conductivity must be an order of 0.1 S/cm, however, the new data in Fig. 5c shows that the conductivity is over 50 S/cm (about 500 times larger). Why is the magnitude of the resistivity (or conductivity) so different between two? The authors should double check the data/analysis and reproducibility of the data, which is a key for publication.
Response: As mentioned previously, the new data are different from the old data (slightly different synthesis route). The reasons for this have already been explained in our last response. We would only like to add that while the reasons for the different resistivity/conductivity values due to different synthesis might be interesting, they do not directly pertain to the main theme of the present manuscript, namely proposing the existence of a TAI phase. We can assure the reviewer that both sets of data, old and new, are entirely reproducible under the synthesis conditions of their respective samples.
The authors mentioned in lines 359-360 as “thermal treatments typically lead to higher density, improved connectivity between adjacent domains, and a reduction in carrier scattering from the grain boundary”. I understand that sintering grains (i.e., thermal treatments) is performed to promote some grain growths from Fig 5a and 5b, and Fig, S6. However, the TEM images in Fig. 5 give me an impression that the density for just pressed sample looks higher than that for sintered assuming that the definition of the density is the ratio between filled area by grains and empty (porous) area in the images. I would agree that the TOTAL connecting area where adjacent grains/domains contact has something to do with the connectivity with respect to conductivity, though it does not look so easy to estimate the area from a 2D image like those in Fig. 5. Have the authors quantitatively estimated it? Otherwise, “improved connectivity” sounds a speculation as it is not obvious.
Furthermore, in general, it is not so simple to precisely estimate the carrier concentration and mobility of porous samples like what the authors have from the Hall data because the orientations and distributions of current flows inside the sample are not trivial against the applied magnetic field due to the presence of randomly oriented chains of grains. I imagined that the samples are more packed, which turns out to be incorrect from the TEM images in the revised version. The Hall data may be also affected by how/where the electrode contacts were made in this case. The authors should provide such information so that readers can judge how reliable the transport measurements were.
Response: There appears to have been a misunderstanding regarding the interpretation of the TEM images. The TEM images are provided, to show the domain size of the nanoparticles of CZTS rather than the sample density. The samples used for mobility (and other transport) measurements are in the form of densely packed pellets. The details of these experimental measurements, including the density of the samples can be found in our previous article (). In fact, the TEM images do not provide any information on the sample density, since they are crushed and dispersed before the imaging. They only provide corroboration for the domain size of the nanoparticles as estimated using XRPD.
In general, the experimental measurements in the present article are provided only to show that they are not in contradiction to the theoretical predictions, as has been stated explicitly, and are not meant as experimental proof of topological surface states. We simply aim to show that the current experimental evidence does not preclude the presence of these surface states, with more rigorous experimental confirmation being part of ongoing work.
Finally, I wonder if topological surface states (TSS) of CZTS grain should vanish at the grain boundaries? When two topologically nontrivial materials are attached, then the original TSSs at the interface should vanish as TSSs could emerge at interface where trivial and nontrivial materials meet. I also wonder if the gapless (metallic) TSSs would be gapped (not metallic) due to interference of the wavefunctions between two sides of the surface of a grain when the size of grains becomes so small (smaller than several nm). If this is the case, what are observed have nothing to do with topological states. It may be prudent to consider other possibilities rather than a contribution from TSSs in this case to provide a conductivity difference between just pressed and sintered samples.
Response: This is an important question which deserves serious consideration; unfortunately however, the answer is unquestionably beyond the scope of this work. The focus in this article is to propose a novel, disorder induced Topological Anderson Insulator phase in a material system. The vast majority of experimental work in the field of topological insulators has been performed on single crystals. The issue of how topological surface states might behave at grain boundaries in nano-polycrystalline materials is not clear. From a theoretical point-of-view, there are some indications that TSS’s from opposite surfaces in these grains might overlap and form hybridized states. While the study of these states is of great interest, it is more appropriately conducted within the context of well-known topological materials such Bi2Se3, rather than a new, disordered phase. We would like to reiterate that the present article is entirely focused on establishing, primarily theoretically, a novel topological phase.
In light of the comments from the present and other reviewers, we have decided to move our limited experimental results into the supplementary information, presenting only the primary theoretical results in the main article.
Reviewer 2 Report
I can see a reasonable improvement of the MS compared to it its early version that I revised couple of weeks ago. The graphics seems to be reasonable, together with the basic science discussed. While an effort has indeed placed to carefully handle my suggestions, there are still a few things that should be taken into account before the acceptance of the article.
- There are feasibility of typos and errors. One such example is this “… are very similar for 100 , The hybrid functional…” There should be a full stop, but not a comma. Similarly, there is an error in this sentence “…by XRD, seFigure 5. Supplementary…” and so on.
- It is not clear what did the authors mean by “The hybrid functional only shifts the conduction bands to a higher energy, justifying our use of the standard, computationally inexpensive PBE functional.” ? Did they perform their band structure calculations using both HSE and PBE?
- Authors have tried show a linear relationship in Fig. 2a. I assume that the data do not really fit to a linear function, as are very much scattered around the regression line. Why is the R2 value not shown? What are the extent of errors involved?
- True disorder in ionic positions is of course rather difficult to simulate within the size-constraints of a DFT supercell with periodic boundary conditions, which impose a long-range order on the system. If this is the case, then what has been discussed should be treated as a “Speculation”?
Author Response
- There are feasibility of typos and errors. One such example is this “… are very similar for 100 , The hybrid functional…” There should be a full stop, but not a comma. Similarly, there is an error in this sentence “…by XRD, seFigure 5. Supplementary…” and so on.
Response: We thank the reviewer for these corrections. We have implemented the changes in the manuscript.
- It is not clear what did the authors mean by “The hybrid functional only shifts the conduction bands to a higher energy, justifying our use of the standard, computationally inexpensive PBE functional.” ? Did they perform their band structure calculations using both HSE and PBE?
Response: The quoted line is from the article by Paier et al (), where the authors show that the hybrid functional only shifts the conduction band to higher energies. As discussed in the article and also our manuscript, using the HSE functional only ‘improves’ the band structure by opening up the band gap. Significantly, the band topology was found to remain the same for both functionals. Therefore, we have made the band structure calculations using only the PBE functional. However, in order to make sure that the underestimation of the gap using the PBE functional does not lead to a false band inversion, we have added a single point, non-selfconsistent calculation using the HSE functional, which confirms the negative band gap, and therefore the band inversion.
- Authors have tried show a linear relationship in Fig. 2a. I assume that the data do not really fit to a linear function, as are very much scattered around the regression line. Why is the R2 value not shown? What are the extent of errors involved?
Response: The reason for the scattering of data points is due to the arbitrary nature of the absolute energies computed with DFT. As seen from Table S1, the total energies of the different configurations, while within a few eV of each other, are nevertheless not exactly the same. This leads to slightly different arbitrary origins of the energy for each configuration. Since the quantities plotted in Fig 2a are differences in energy (band inversion strength vs crystal field splitting), certain fluctuations are to be expected. In general, these deviations do not seriously affect the results. The Pearson’s r coefficient for the data is explicitly stated as 0.84 in the text below the figure in the manuscript. Based on this value, we can assert that there is a sufficiently good positive linear relationship between the two variables.
As for errors, it is obvious that these numbers are differences between calculated energy levels from band structures. Being theoretical non-stochastic data, therefore, the idea of errors does not apply.
- True disorder in ionic positions is of course rather difficult to simulate within the size-constraints of a DFT supercell with periodic boundary conditions, which impose a long-range order on the system. If this is the case, then what has been discussed should be treated as a “Speculation”?
Response: While long-range order is indeed imposed on all DFT calculations within the supercell formalism, the effects of this order can in general be ignored in the absence of long-range Coulomb interactions. Since our system is charge-neutral, long-range interactions can be safely disregarded. For the disordered CZTS system, interactions are therefore short range interactions between electrons at each lattice site, and a mean field accounting for their first, maybe second nearest neighbors. In fact, this assumption is routinely made in DFT calculations of systems which are in some way inherently aperiodic – for example, crystalline defects, vacancies, or surfaces. What is important to ensure is that the supercell is large enough to ignore interactions between an electron and its periodic copy. For systems which are not strongly correlated (e-e correlation is not considered in DFT), this generally corresponds to a separation of at least ~10 angstroms, depending on the nature of the system.
For our given problem, it is important to note that the TAI behavior is ultimately determined by the on-site and hopping terms in the Hamiltonian. While the onsite energy represents the energy of the electron in a particular orbital in the presence of other ions, the hopping integral instead measures the frequency of transition of the electron between adjacent orbitals. Specifically, the cation disorder gives rise to a highly random, long-range, on-site disorder potential, which, as explained in the article, leads to a renormalization of the effective band mass leading to a negative value, which leads to a TAI phase. This mechanism is thus solely determined by the effective/mean field due to the full cation disorder. Should the long-range periodicity of the DFT supercells have significantly affected the on-site energies, it would have destroyed the aperiodic nature of the disorder potential, suppressing the manifestation of TAI behavior. Long-range periodicity is in fact known to be one of the conditions negating the presence of a TAI phase. As such, we can confidently assert that the manifestation of a TAI phase indicates that the system is sufficiently aperiodic, and the long-range order of the DFT supercell can safely be ignored.
Reviewer 3 Report
This manuscript describes a method to prepare a topological Anderson insulator Cu2ZnSnS4. By using density function theory calculations, the authors investigated the order-disorder transition of the Cu2ZnSnS4 by introducing cation randomization. The authors carefully examined the valence bands in different randomized states. With these theoretical calculations, the authors conclude the possibility to prepare the topological insulator. Then, the prepared samples were analyzed with good characterization. There are still some important issues before publication indicated below:
- There were too many marks in the manuscript. It is so difficult to read. Please remove these marks or un-necessary font colors.
- Please provide marks for each atom for the crystal structures in Figure 1. Although it was explained in the caption, it would be easier for readers to understand with symbols.
- It is not clear how the authors perform the randomization in each plane in Figure 1. We only know the stoichiometry was kept but how the atoms were swapped or exchanged? Please provide a more specific description on the randomization method.
- In figure 5, was the “just-pressed” sample the original sample before thermally heating? It seems to me the authors were comparing the particle size before and after sintering. Please revise the caption and related words in the text.
- In section 3.3, it is not quite convincing that these synthesized materials were topological insulators. More experimental investigation should be provided. Moreover, how can one verify the disorder state in experiments as those in the theoretical calculations?
Author Response
There were too many marks in the manuscript. It is so difficult to read. Please remove these marks or un-necessary font colors.
Response: It appears that these annotations were introduced during the editorial process. As far as we are aware, we submitted a clean manuscript for the peer review process.
Please provide marks for each atom for the crystal structures in Figure 1. Although it was explained in the caption, it would be easier for readers to understand with symbols.
Response: We thank the reviewer for this comment. We have updated Fig 1 accordingly.
It is not clear how the authors perform the randomization in each plane in Figure 1. We only know the stoichiometry was kept but how the atoms were swapped or exchanged? Please provide a more specific description on the randomization method.
Response: The randomization was performed in a very straightforward manner. A total of 10 different configurations were generated (see Table S1), and the system with the lowest energy was presented in the main article. It should be noted that the energy difference between the different configurations is of the order of a only few eV, so they might be considered more or less isoenergetic. We have added a short explanation in the new computational methodology section.
In figure 5, was the “just-pressed” sample the original sample before thermally heating? It seems to me the authors were comparing the particle size before and after sintering. Please revise the caption and related words in the text.
Response: As the reviewer correctly understands, the “just-pressed” sample refers to the sample before thermal treatment. We have accordingly revised the caption and the text.
In section 3.3, it is not quite convincing that these synthesized materials were topological insulators. More experimental investigation should be provided. Moreover, how can one verify the disorder state in experiments as those in the theoretical calculations?
Response: We would like to point out that nowhere in the manuscript is it stated that the experimental results prove the presence of a topological insulator. We simply argue that the experimental results do not disagree with what is seen from the calculations: a topological insulator should show a higher conductivity with a higher surface-to-volume ratio. As we have explicitly stated in the manuscript, the majority of experimental characterization of topological surface states have to date been performed with ARPES, on single crystal samples. Studies on topological insulators in the nano-polycrystalline phase are at the nascent stage, and as far as we are aware, there are no clear, go-to experimental methodologies which can “prove” the presence of topological surface states in such a phase. More sophisticated transport measurements are ongoing, as clearly mentioned in the conclusion of the manuscript.
The verification of the disorder state in the samples, on the other hand, is straightforward using X-ray powder diffraction. This disorder state has been extensively characterized in previous work, to which we refer the reviewer. In short, the ordered form of CZTS has a tetragonal space group (I-4), whereas the current samples, synthesized using high energy reactive mechanical alloying (ball milling), lead to and X-ray diffraction pattern corresponding to a cubic space group (F-43m) (see supplementary information). This is actually the well-known ZnS (sphalerite) structure, with a higher symmetry than the tetragonal structure. Given the presence of different cations in stoichiometric proportion, and the sphalerite diffraction pattern, it can be concluded that the cations are randomly and uniformly distributed at the cation lattice sites, instead of in alternating, ordered Cu-Zn and Cu-Sn layers as in the ordered, tetragonal form.
In light of the comments from the present and other reviewers, we have decided to move our limited experimental results into the supplementary information, presenting only the primary theoretical results in the main article.
Reviewer 4 Report
The authors report on a candidate material as an Anderson topological insulator (a TI born from increased disorder). They perform calculations of the bulk band structure and find some support for band inversion at certain locations in the Brillion zone. They also provide some support for the presence of surface states with further calculations of the surface band structure. Finally measurements on the material are performed and the electrical properties are reported.
I think the presentation of the results are fine, but the experimental results are very weak as support for the presence of surface states. I do not know how the authors have made the conclusion that the grain size would necessarily give information about the presence of surface states in this material. How much does the grain size change after being pressed? How do surface states from different grains couple together? If the sample was simply metallic grains that were pressed together tightly, the conductivity would increase. I think the calculations are valid as they are, they can be reported and perhaps many will find this information useful, but the support from experiment is just not conveyed in this report. I do not recommend publication of this manuscript in its present form. Perhaps the authors can report solely on the calculation results which may be useful.
Author Response
I think the presentation of the results are fine, but the experimental results are very weak as support for the presence of surface states. I do not know how the authors have made the conclusion that the grain size would necessarily give information about the presence of surface states in this material.
Response: We would like to point out that nowhere in the manuscript is it stated that the experimental results prove the presence of a topological surface states. We simply argue that the experimental results do not disagree with what is seen from the calculations: a topological insulator should show a higher conductivity with a higher surface-to-volume ratio. As far as this concerns nano-polycrystalline samples, we argue that grains with larger domain sizes should have a higher surface-to-volume ratio, and as such, more topological surface states, indicating higher mobility, which is what we see in our samples. We present this as a necessary, not sufficient, condition for the presence of topological surface states.
How much does the grain size change after being pressed? How do surface states from different grains couple together? If the sample was simply metallic grains that were pressed together tightly, the conductivity would increase. I think the calculations are valid as they are, they can be reported and perhaps many will find this information useful, but the support from experiment is just not conveyed in this report. I do not recommend publication of this manuscript in its present form. Perhaps the authors can report solely on the calculation results which may be useful.
Response: The change in the domain size of the grains is clearly shown in both the TEM images, and also from the XRPD results in the supplementary information.
How surface states from different grains couple together is an interesting question, albeit one which falls outside the scope of the present article. The focus in this article is to propose a novel, disorder induced Topological Anderson Insulator phase in a material system.
The vast majority of experimental work in the field of topological insulators has been performed on single crystals. The issue of how topological surface states might behave at grain boundaries in nano-polycrystalline materials is not clear. From a theoretical point-of-view, there are some indications that TSS’s from opposite surfaces in these grains might overlap and form hybridized states. While the study of these states is of great interest, it is more appropriately conducted within the context of well-known topological materials such Bi2Se3, rather than a new, disordered phase. We would like to reiterate that the present article is entirely focused on establishing, primarily theoretically, a novel topological phase.
In light of the comments from the present and other reviewers, we have decided to move our limited experimental results into the supplementary information, presenting only the primary theoretical results in the main article.
Round 2
Reviewer 1 Report
The revised manuscript looks an almost purely theoretical work as the authors quite weakened their assertion about a successful demonstration of experimental evidence for the existence of TAIs. As the authors mention in discussion and conclusions, further transport measurements are necessary, which sounds reasonable.
Discussion part became slightly too short against other chapters. I am not sure what the authors can/should do, but is it a good idea to combine results and discussion sections? Anyhow, I would recommend the authors reconsider the structure/balance between sections of the manuscript.
Author Response
We thank the reviewer for their helpful comments. Following their suggestions, we have split the shortened "discussion" section, and merged it partially with the "results" (now Results and discussion), and "conclusions" section.
Reviewer 3 Report
The authors have revised the manuscript properly with reasonable modification. It is now in a good shape for publication.
Author Response
We thank the reviewer for their helpful suggestions, and for recommending our manuscript for publication.
Reviewer 4 Report
I think moving the experimental results to the supplemental material was the best solution. The text also better explains that the results are supportive but more studies are required. I would publish the manuscript in its current form.
Author Response
We thank the reviewer for their useful suggestions, and for recommending the publication of our manuscript.
This manuscript is a resubmission of an earlier submission. The following is a list of the peer review reports and author responses from that submission.
Round 1
Reviewer 1 Report
The authors are going to report that Cu2ZnSnS4 (CZTS) is the first candidate for a disorder-induced Topological Anderson Insulator in a real material system using ab initio calculation and transport measurements for the resistivity and the impedance spectroscopy of CZTS samples. The authors clearly mention the motivation and importance of their research in the manuscript. Unfortunately, it seems that the CZTS is not suitable for spintronics. However, this is a topical issue and, indeed, it is very important to deepen our understanding how to induce topological phases of matter for their feasible applications to future electronics, though. Thus, I would like to recommend considering this manuscript as a possible publication in Nanomaterials after the authors reply to my concerns as written below.
First, I don’t understand why the authors conclude that their CZTS samples show surface dominated conduction. It sounds too naive to consider that their samples are sufficiently bulk insulating only from their high resistivity. Because the resistivity can go high if the mobility is very low. In addition, the temperature dependence of the resistivity of their samples (Fig. 5a) could be insulating or semiconducting. If the main contribution of the resistivity is due to metallic surface states, then the states cannot explain the observed temperature dependence, i.e., the resistivity should go up when temperature increases. The authors should elaborate more on scientific evidence for the surface dominated conduction. It could be useful to measure the Hall resistivity to see the number of conduction channels, mobility, and the carrier densities for each channel.
Secondly, the frequency dependence of the imaginary part of the impedance Z in Fig. 5d looks more inductive (wL + higher order of w) rather than capacitive (1/wC). I don’t think that the equivalent circuit for a series of parallel circuit of a resistor and a capacitor in the bulk of the samples illustrated in the inset in panel of Fig. S8 cannot explain the frequency dependence in Fig. 5d. The authors should demonstrate that the presence of metallic surface/interface states on crystals in multi polymorphs give rise to an inductive behavior as a function of frequency.
Thirdly, the authors should demonstrate how a Nyquist plot in Fig. S8 can be obtained from the data. I assumed that the data in Fig. 5d are used but the magnitude of the imaginary part of the impedance in Fig. S8 (about 1 ohm) is very different from that in Fig. 5d (from 0 to a few ohm). If the authors used another set of the data, those should be disclosed.
Finally, two miner comments:
The “TAI’s have …” in the line 78 should be just “TAIs have”.
The figure caption of Fig. 4 misses explanations for 4(d) and 4(f).
Reviewer 2 Report
The manuscript ttiled „Topological Anderson Insulator in Cation-Disordered 2 Cu2ZnSnS4" by B. Mukherjee et al. shows results of electrical transport characterization and DFT calculations of electronic structure of Cu2ZnSnS4 samples made with different routes in order to obtain different structural parameters of the resultant samples. The results of the DFT calculations present a beautiful world of computational virtual crystals in which you can observe effects related to the presence of sophisticated band arrangment as a consequence of differences in lattice type. The second part of the paper presents results obtained for a low quality materials. Unfortunately I do not see absolutely any signatures of the presence of topological states in these results. Therefore the experimental part of the manuscript falls away from the DFT calculations and therefore the present manuscript does not present a unified picture of this material. I cannot recommend publication of this manuscript in Nanomaterials.
Reviewer 3 Report
This work may be interesting from a fundamental point of view. The basic science discussed is OK, but it is not a well organized and well-written ms. Therefore, I ask the authors to revise their ms based on my comments given below.
- The standard of English writing is no good. The authors should devote their time to produce a credible draft that can be easily understood by readers of the journal. One such example could be this sentence: In a recent study, Isotta et al[29] demonstrated remarkably improved thermoelectric properties in a cubic polymorph of CZTS with complete cation disorder, synthesized using high energy reactive ball-milling, which shows a simultaneous improvement of both the Seebeck coefficient and electrical conductivity, as well as a lower thermal conductivity compared to the ordered tetragonal polymorph. Using DFT band structure calculations, we argue that introducing full cation disorder in CZTS drives it into a TAI phase; experimental measurements of resistivity and impedance spectroscopy are in agreement with the surface-dominated transport which such a phase is expected to host. In fact, this sentence reads like a paragraph. While I am reading the last line of the sentence, I almost forgot what they wrote in the first line of the same sentence. It is, of course, a dull writing. I hardly see such writings in any standard international journal. As it seems, the authors of the study are non-native, but this work can be interesting after a careful rewriting of the work.
- The objective of the paper is written in the last three lines of the introduction section, while the introduction is shown to be too long. Such a presentation is tedious, as the audience cannot quickly grasp what the authors of the work intended to say!
- Background references are missing in many places. For instance, this following sentence needs references. It should be noted that the GGA tends to underestimate the band gap for most compounds, which may be corrected using computationally expensive hybrid functionals.
- Why did they use an energy cutoff of 300 eV? In general, it should be over 400 eV!
- Why is the PBE functional chosen, even though the authors demonstrate about the use of the hybrid functionals “However, a hybrid Hartree-Fock/DFT study[33] has established that the band topology for both computational schemes (hybrid and PBE) are very similar, with the hybrid functional shifting the conduction bands to a higher energy.”
- Why different k-point grids were being used for geometry optimization, and scf calculation, etc?
- Discussion and conclusion section should be split as Discussion and Conclusion. It is messy.